# Effects and Mechanism of Micro-Drilling Parameters on the Drilling Force and Hole Morphology of *Sapindus mukorossi* Seeds

Suxiao Zhao [1,2], Xiaopeng Bai [1,2,*], Daochun Xu [1,2,*] and Wan Cao [1]

1   School of Technology, Beijing Forestry University, Beijing 100083, China
2   Laboratory of State Forestry Administration on Forestry Equipment and Automation, No. 35 Tsinghua East Road, Haidian District, Beijing 100083, China
*   Correspondence: xiaopengbai@bjfu.edu.cn (X.B.); xudaochun@bjfu.edu.cn (D.X.)

**Abstract:** *Sapindus mukorossi* seeds are commonly drilled and processed into Buddha beads and other craft products. To improve the machining qualities of the micro-drilling of *S. mukorossi* seeds, the effects of drilling parameters (feed rate, drilling speed, and drill diameter) on the drilling force, hole diameter and hole morphology were found. The mechanism behind these changes was further analyzed. The drilling parameters and machining process for the micro-drilling of *S. mukorossi* seeds were optimized. Results indicated that each drilling parameter influenced the drilling force and hole-machining quality by affecting the amount of cutting, the working conditions of the drill bit, chip formation, and the generation of drilling heat. Moreover, the feed rate and drill diameter were the main factors that influenced the drilling force, and the drilling speed played a leading role that influenced hole-machining quality. Interestingly, for the drilling of *S. mukorossi* seeds, a feed rate of 0.03 mm/r, a drilling speed of 6 m/min, and a drill-bit diameter of 0.9 or 1.5 mm resulted in a hole with high accuracy, good hole morphology, and minimal wear on the drill bit. This study facilitates the selection of suitable drilling parameters and processing technology for the micro-drilling of *S. mukorossi* seeds to improve processing accuracies and qualities, and provides a theoretical basis for the improvement of related drilling processing equipment.

**Keywords:** *Sapindus mukorossi*; micro-drilling; drilling parameters; drilling force; hole morphology

## 1. Introduction

Seeds have played, and continue to play, an important role in human culture as one of the earliest decorative materials. Since ancient times, people have commonly used hard-shelled, richly colored seeds as beads and drilled holes through them to make them into necklaces, bracelets, and other decorative objects [1]. Currently, one of the most common seed crafts are Buddha beads (which have an important place in Buddhism as a counting device for chanting). The seeds used to make Buddha beads are called bodhi seeds. A fixed number of bodhi seeds is drilled and strung into strings to make Buddhist prayer beads and bracelets to be worn on the hands [2]. However, the small size and hard shell of most seeds makes it challenging to produce holes on them.

Owing to its high material removal rate and wide applicability, micro-drilling is the most cost-effective method for producing holes in woody materials, especially seeds [3]. For micro-drilling technology, people have conducted a lot of research on metal, non-metal, organic materials and composite materials from different angles [4–9]. However, for woody materials, micro-drilling has not seen much attention, where it has mainly been used as an inspection technique. An example of this is the micro-drilling resistance method for wood (which is a micro-damage inspection technique) [10–12]. While research on the process parameters of micro-drilling of woody materials, such as *S. mukorossi* seeds, is lacking, a review of the literature shows that there is little research on the changes in drilling forces

and hole morphology in wood materials under different drilling parameters, and that the influence of drilling forces on the drilling process has not yet been revealed. The correct selection of drilling parameters is of great importance for improving the drilling efficiency, ensuring durability of the drilling tool, reducing costs, and improving quality [13]. In addition, drilling a round hole in such a tiny and hard structure as a seed is not easy. The layered structure of seeds leads to new problems in the drilling process, such as tool wear and roughness of the hole appearance. Therefore, the design of relevant equipment (e.g., special drilling machines for seeds) lacks experimentally obtained parameters.

To address this knowledge gap, this study conducted drilling experiments on *S. mukorossi* seeds, which are the most common bodhi seeds [14]. *S. mukorossi* is mainly found in Southeast Asia, and North and South America. It is an important biological resource for forestry [15], and is widely used in medicine [16], as an ingredient in cleaning products [17], in the production of biodiesel [18], and as a bead accessory. *S. mukorossi* fruits (Figure 1a) consist of a translucent yellow coating and black seeds (Figure 1b). *S. mukorossi* seeds consist of a seed shell (which is a hard, dense kernel coating about 2 mm thick) and a seed kernel (consisting of 42.3% oil, including various higher fatty acids) (Figure 1c). Because of their hard outer shell, *S. mukorossi* seeds are commonly drilled and processed into Buddha beads (Figure 1d) and other handicrafts.

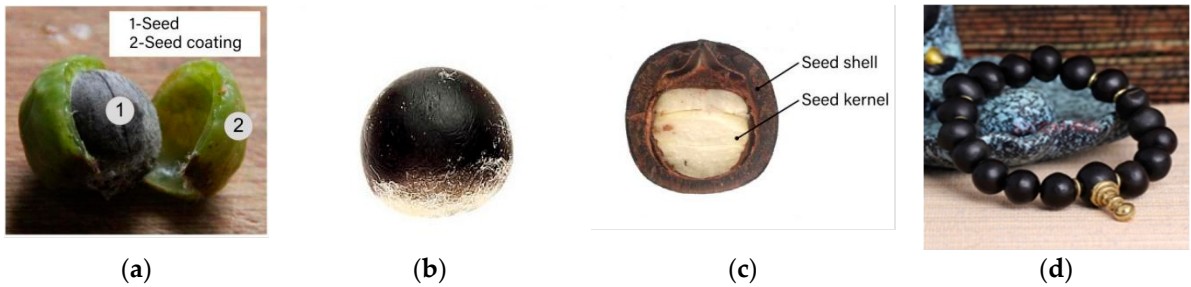

**Figure 1.** *S. mukorossi*: (**a**) fruit, (**b**) seed, (**c**) seed structure, and (**d**) Buddhist prayer beads.

This study focuses on the drilling parameters for the micro-drilling of *S. mukorossi*. Specifically, the effects of feed rate, drilling speed, and drill diameter on the drilling force, hole diameter, and hole morphology are analyzed. The optimal selection of suitable drilling parameters for the micro-drilling processing of *S. mukorossi* seeds (and similar wood structures) to improve the processability and accuracy of the drilling process are determined. A theoretical basis is also provided for the improvement of drilling equipment. These findings are also useful in related micro-drilling applications in wood materials (e.g., micro-damage inspection technology).

## 2. Material and Methods

### 2.1. Seed Materials

High-quality *S. mukorossi* trees in southeastern China were selected as the source of seeds. The selected seed material was obtained in October 2022 in Fujian Province, China (26°14′36″ N; 117°35′48″ E). A sufficient number of *S. mukorossi* fruits of moderate size, regular shape, and uniform quality were obtained from *S. mukorossi* trees that had a high fruiting rate, full fruit, few seed pests, and minimal disease. The *S. mukorossi* fruits were dried naturally in a shady place and peeled off. The moisture content of the *S. mukorossi* seeds was 5.58%, and its percentage was calculated on a wet weight basis. Processed seeds were stored in airtight plastic bags in the refrigerator at 4 °C for further study.

### 2.2. Experimental Design

Fixed seeds of the same size were drilled with a micro-drill with the feed rate, drilling speed, and drill diameter as independent variables. To reduce experimental error, 64 homogeneous *S. mukorossi* seeds of approximately 12 mm diameter were selected, 12 of which were used as test pieces and spare samples (a replacement job in case of

drilling errors such as operational errors or drilling parameters being incorrect) for the drilling process. The range of drilling parameters for the drilling experiments was determined by preliminary test drilling. With a feed rate of 0.02–0.07 mm/r, a drilling speed of 3–24 m/min, and a drill diameter of 0.9–1.5 mm, 36 sets of experiments were designed for 36 *S. mukorossi* seeds. The drilling parameters for these experiments are shown in Table 1. In addition, 6 *S. mukorossi* seeds were used to repeat the drilling experiment under optimal drilling parameters, and the resulting hole morphology was observed.

**Table 1.** Drilling parameters of the drills.

| No. | *D* (mm) | *U* (mm/r) | *v* (m/min) | No. | *D* (mm) | *U* (mm/r) | *v* (m/min) | No. | *D* (mm) | *U* (mm/r) | *v* (m/min) |
|-----|----------|------------|-------------|-----|----------|------------|-------------|-----|----------|------------|-------------|
| 1 | 1.0 | 0.02 | 24 | 13 | 1.0 | 0.03 | 12 | 25 | 0.9 | 0.03 | 6 |
| 2 | 1.0 | 0.02 | 21 | 14 | 1.0 | 0.03 | 9 | 26 | 1.3 | 0.03 | 6 |
| 3 | 1.0 | 0.02 | 18 | 15 | 1.0 | 0.03 | 6 | 27 | 1.5 | 0.03 | 6 |
| 4 | 1.0 | 0.02 | 15 | 16 | 1.0 | 0.03 | 3 | 28 | 0.9 | 0.03 | 12 |
| 5 | 1.0 | 0.02 | 12 | 17 | 1.0 | 0.04 | 12 | 29 | 1.3 | 0.03 | 12 |
| 6 | 1.0 | 0.02 | 9 | 18 | 1.0 | 0.05 | 12 | 30 | 1.5 | 0.03 | 12 |
| 7 | 1.0 | 0.02 | 6 | 19 | 1.0 | 0.06 | 12 | 31 | 0.9 | 0.03 | 18 |
| 8 | 1.0 | 0.02 | 3 | 20 | 1.0 | 0.07 | 12 | 32 | 1.3 | 0.03 | 18 |
| 9 | 1.0 | 0.03 | 24 | 21 | 1.0 | 0.04 | 6 | 33 | 1.5 | 0.03 | 18 |
| 10 | 1.0 | 0.03 | 21 | 22 | 1.0 | 0.05 | 6 | 34 | 0.9 | 0.03 | 24 |
| 11 | 1.0 | 0.03 | 18 | 23 | 1.0 | 0.06 | 6 | 35 | 1.3 | 0.03 | 24 |
| 12 | 1.0 | 0.03 | 15 | 24 | 1.0 | 0.07 | 6 | 36 | 1.5 | 0.03 | 24 |

In the experiments, the drilling positions were divided into seven groups according to the path of the drill and the structure of the *S. mukorossi* seeds (as shown in Figure 2). The drilling forces for each drilling position within the seeds and the diameters of the inlet and outlet of each hole were collected, and the relationship between each drilling parameter and the drilling force and hole diameter was analyzed. Each group of parameter combinations was repeated three times. The average of the three replicates was calculated and is shown in the chart.

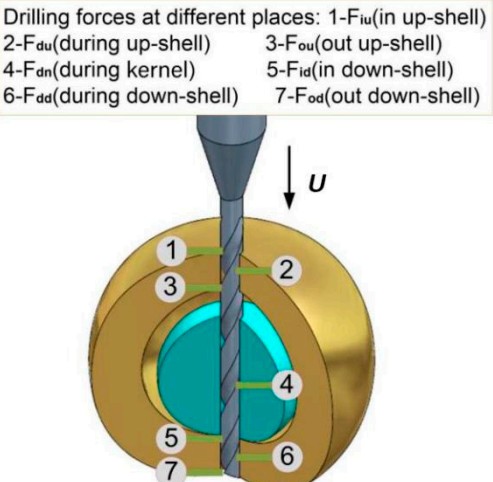

**Figure 2.** Stages of the drilling process.

### 2.3. Drilling Experiment

Experiments were performed for dry drilling at low speed. The experimental platform (Sauer Ultrasonic 50, DMG, Bielefeld, Germany) is shown in Figure 3. The drills, which were made by UNION TOOL (Union Tool Co., Ltd., Shanghai, China), all belonged to the lengthening blade series and had two blades made from carbide materials. The helix and tip angles were 30° and 130°, respectively. Because of drill wear, the drills were replaced after every two drillings.

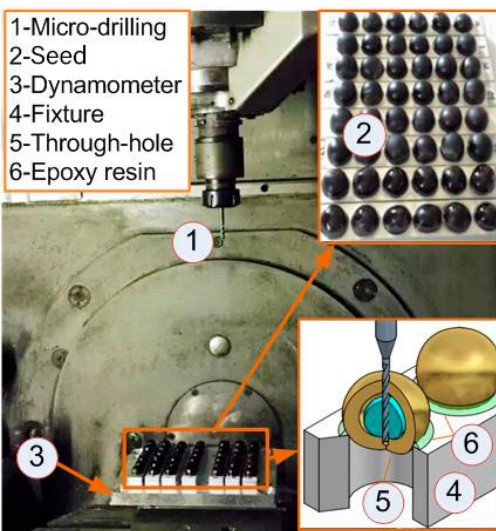

**Figure 3.** Experimental platform of drillings.

Owing to the smooth and hard spheres of *S. mukorossi* seeds, the seed was difficult to fix during drilling. To avoid skewing of the hole axis, enlargement of the hole diameter, and deformation of the round hole shape, the seeds were glued with a drilling die by epoxy resin. The umbilicus of all seeds was placed perpendicular to the horizontal plane. This ensured the accuracy and stability of the drilling process.

### 2.4. Measurement and Observation

A dynamometer (Kistler 9257B) recorded the drilling force with high accuracy (the z-direction sensitivity was $-3.7$ pC/N) and high repeatability. The Dynoware data acquisition system (2825A-02 V2.6.3.12), which is a general-purpose data acquisition and analysis software supporting the Kistler dynamometer, was adopted to measure the micro-drilling forces (the sampling rate of the drilling force was 1000 Hz). The measured drilling force ($F_z$) was collected and analyzed. All test data were taken during a phase of stable cutting (drilling without environmental factor interference).

Scanning electron microscopy (SEM) was performed to evaluate the structure morphology of seed samples. A sputtering coater (IB-3, Eiko, Yokohama, Japan) sputter-coated the layer of broken seed samples with gold for 300 s, and an SEM system (Merlin, Zeiss, Oberkochen, Germany) belonging to the National Center for Electron Microscopy in Beijing at Tsinghua University imaged the samples. The outer side of the seed shell, middle part of the seed shell, inner side of the seed shell, and seed kernel were observed and photographed through SEM at 2000× magnification. The detailed test procedures were presented in our previous research [19,20].

A Trinocular Camera Stereo Microscope (XZT-CT, Beijing Shangliuguang Instrument Co., Ltd., Beijing, China) provides high-definition images which were used to observe the morphology of the holes in the seeds. A UV-G particle-size-analysis microscope was used to accurately measure the hole diameter.

## 3. Results and Discussion

### 3.1. Structural Analysis of Different Positions within the Seeds

When the drill diameter, feed rate and drilling speed were constant, the drilling force varied among the different positions on the *S. mukorossi* seeds because of their complex internal structure. The internal structure of the *S. mukorossi* seeds was investigated by SEM (scanning electron microscopy). Figure 4 shows images of internal structures of seeds magnified 2000 times. The seed shell is roughly divided into three layers from the outside to the inside. The outer side of the seed shell is a continuous dense layer with alternating undulation, which has good toughness and strength (Figure 4a). The middle part of the seed

shell is smoother than the outer side of the seed shell, but its surface is evenly distributed with micron-level cracks; its toughness is relatively poor (Figure 4b). The inner side of the seed shell is rougher and can be observed with irregularly sized scaly processes and a large number of hole structures, which are more prone to rupture (Figure 4c). For the seed kernel, observations show that the kernel has a loose porous structure, which makes it most easily broken during drilling (Figure 4d).

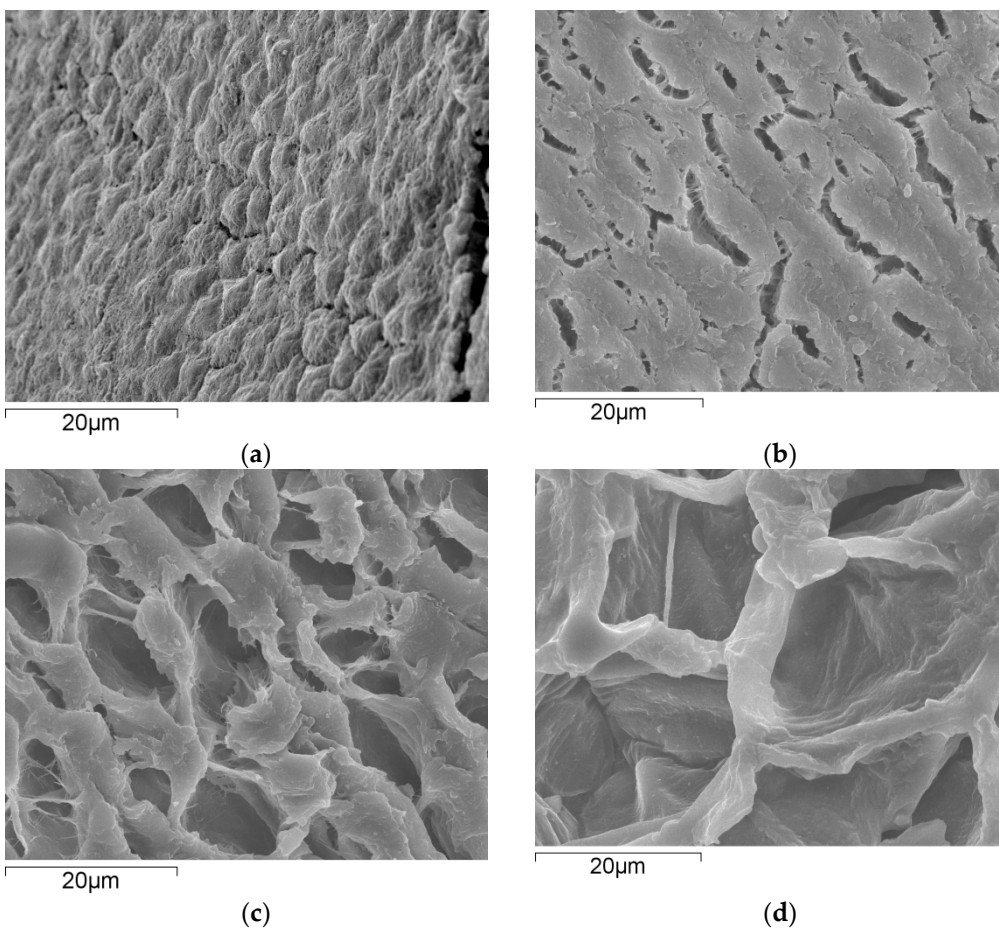

**Figure 4.** The internal structure of the *S. mukorossi* seeds. (**a**) Outer side of the seed shell, (**b**) middle part of the seed shell, (**c**) inner side of the seed shell, and (**d**) seed kernel.

### 3.2. Effect of Feed Rate on Drilling Force and Hole Diameter

Figure 5a–d show the relationship between the feed rate ($U$) and the drilling force ($F_z$) generated at different positions in the hole when the drill diameter ($D$) and drilling speed ($V$) were constant. For a drill diameter of 1.0 mm, a drilling speed of 6 m/min, and a feed rate of 0.02–0.07 mm/r, the drilling force at different positions of the *S. mukorossi* seeds was up to 25 N. When the drilling speed was 12 m/min and the feed rate was in the range of 0.02–0.07 mm/r, the drilling forces at different positions of the seeds were up to 16 N. For a fixed drill diameter (1.0 mm) and drilling speed (6 m/min or 12 m/min), the drilling force increased with increasing feed rate at all positions (except kernel) on the seeds, except for the special case of $V$ = 12 m/min and $U$ = 0.05–0.07 mm/r (Figure 5d). This phenomenon can be explained by the following reasons. First, an increase in feed rate leads to an increase in the cutting thickness and cutting area, which increases the drilling force [21]. Second, *S. mukorossi* seeds have high plasticity. Thus, a greater feed rate will lead to greater plastic deformation, increasing the friction coefficient between the chip and the front tool surface, which increases the drilling force. Furthermore, an increase in feed rate increases the height of the residual drilling area, resulting in a rougher hole wall [22] and an increase in friction with the sub-rear tool face of the guiding part of the drill, which also serves to increase the

drilling force. In addition, a higher feed rate leads to more material being removed. This increases the torque and drilling power [23], in turn increasing the drilling heat generated per unit time and a rise in drilling temperature. The higher temperature accelerates the wear of the carbide drilling tool and reduces drilling capacity, thus further increasing the drilling force. Additionally, because the kernel structure is loose and easily broken, the change in drilling force with different feed rate is not obvious (Figure 5b).

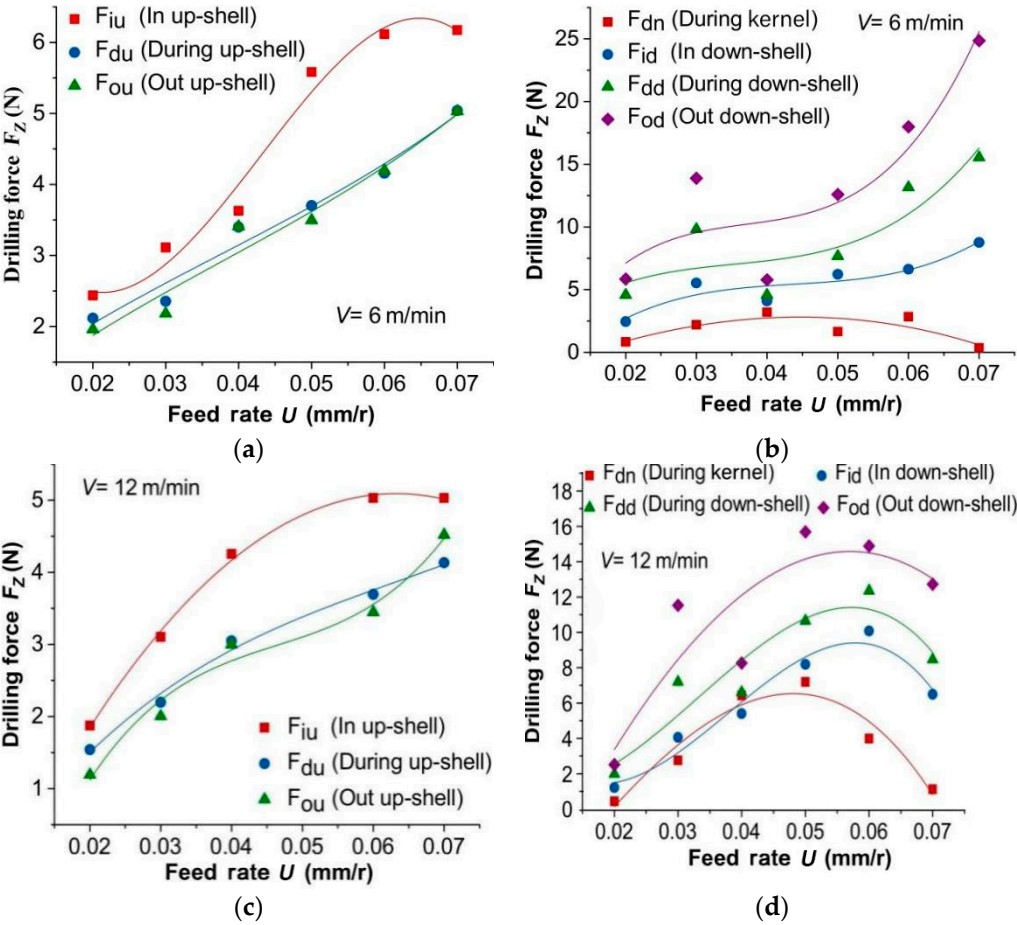

**Figure 5.** Drilling forces as a function of feed rate. (**a**,**b**) The relationship between $U$ and $F_z$ when $D = 1.0$ mm and $V = 6$ m/min, (**c**,**d**) The relationship between $U$ and $F_z$ when $D = 1.0$ mm and $V = 12$ m/min.

For a drilling speed of 12 m/min and a feed rate of 0.05–0.07 mm/r, the drilling force generated during drilling through the seed kernel and the down-shell gradually decreased with increasing feed rate, as shown in Figure 5d. The reason for this is that when drilling at higher speeds, a larger feed rate facilitates the removal of the wood fiber from the seed kernel and down-shell, which reduces the deformation resistance of the material during drilling and reduces the drilling force. In addition, at higher drilling speeds, a higher feed rate for drilling *S. mukorossi* seeds is beneficial for the formation of good chips and reduces the frictional resistance between the chips and the front tool face, resulting in a reduction in the drilling force.

Figure 6 shows the relationship between the feed rate and diameters ($D'$) of the inlet and outlet holes when the drill diameter and drilling speed were constant. For a drill diameter of 1.0 mm, drilling speed of 12 m/min, and feed rate of 0.02–0.07 mm/r, the inlet and outlet diameters of the seed holes varied in the range of 0.8–1.0 mm. When the feed rate was less than 0.05 mm/r, the hole diameter gradually decreased with increasing feed rate. There are two reasons for this. First, an increase in feed rate increases the height of the residual drilling area. Second, an increase in drilling force increases the vibration

amplitude of the drill bit, leading to the rupture of the wood-fiber structure of the seed shell and an increase in the burr of the hole wall, which affects the processing accuracy of the hole [24]. When the feed rate was greater than 0.05 mm/r, the hole diameter gradually increased with an increasing feed rate. This is because a higher feed rate increased the drilling force, leading to an increase in the degree of bending and deformation of the drill bit, and deflection of the drill bit, which enlarged the hole.

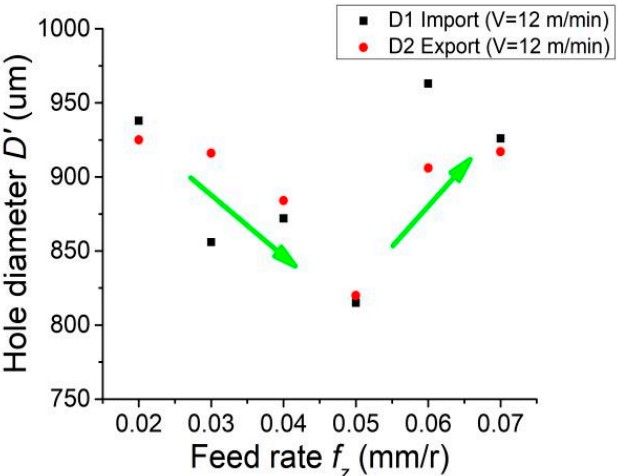

**Figure 6.** Hole diameter against feed rate. The relationship between $U$ and $D'$ when $D = 1.0$ mm and $V = 12$ m/min.

This analysis reveals that to reduce the drilling force and avoid undesirable problems (such as low processing accuracy of the hole and high surface roughness due to drill-lead deflection and workpiece vibration), the feed rate should be as low as possible. However, considering the production efficiency and to prevent drilling heat from affecting the durability of the drill and the quality of the hole, the feed rate should be increased appropriately to shorten the drilling time and promote the dissipation of drilling heat. The recommended feed rate is 0.03 mm/r.

### 3.3. Effect of Drilling Speed on the Drilling Force and Hole Diameter

Figure 7a,b shows the relationship between the drilling speed and the drilling force generated at different positions in the hole when the drill diameter and feed rate were constant. For a drill diameter of 1.0 mm, feed rate of 0.03 mm/r, and drilling speed of 3–24 m/min, the drilling force was up to 15 N. The drilling force decreased with increasing drilling speed at all positions (except kernel) on the seeds, which we explain as follows. First, a higher drilling speed facilitates the drilling and removal of wood-fiber structure from the seeds. This forms good chips and reduces the coefficient of friction between the chips and the front tool face, resulting in a reduction in the drilling force. Second, good chip formation is beneficial in reducing the roughness of the hole wall; a lower surface roughness of the hole wall reduces the frictional force on the guiding part of the drill bit, which serves to reduce the drilling force [25]. Furthermore, the reduction in friction on the hole wall helps to reduce the generation of drilling heat, and the formation of good chips helps to dissipate the drilling heat. These effects reduce the drilling temperature and the wear of the drill bit, thus reducing the drilling force [26]. However, when drilling at high speeds, the drilling temperature will increase, accelerating tool wear and increasing the drilling force. In Figure 7, the decreasing trend of drilling force becomes slower as the drilling speed increases. In addition, because the kernel structure is loose and easily broken, the change in drilling force with a different feed rate is not obvious. However, it still showed a trend of first increasing and then decreasing with an increase in drilling speed. This is because with an increase in drilling rate, the vibration amplitude of the drill bit increases, leading to an increase in friction between the sub-rear face of the guide section

and the hole wall (seed shell), resulting in an increase in drilling force. However, the higher drilling rate is conducive to removing the wood-fiber structure of the seed and forming good chips, reducing the friction coefficient between the chips and the front cutter face, thus reducing the drilling force. (Figure 7b).

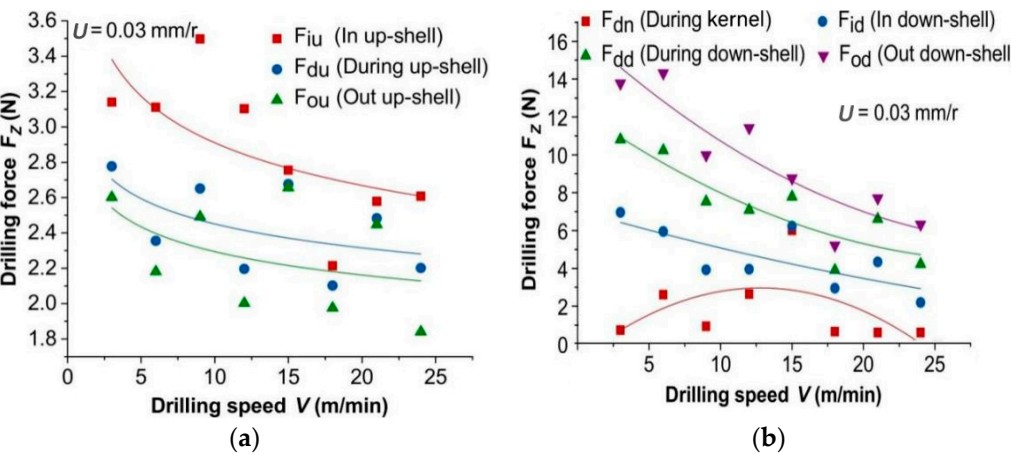

**Figure 7.** Drilling force as a function of drilling speed. (**a**,**b**) When $D = 1.0$ mm, $U = 0.03$ mm/r, the relationship between $V$ and $F_z$.

Interestingly, when the bit diameter, feed rate, and drilling speed were constant, the drilling force at the down-shell was greater than the drilling force at the up-shell, which was greater than that at the seed kernel (as can be seen from Figure 7). Moreover, the force decreased when drilling into the up-shell and increased when drilling out of the down-shell. The reason for this is that the *S. mukorossi* seeds consist of a hard seed shell (Figure 4a–c) and a loosely structured seed kernel (Figure 4d), which is softer than the seed shell, such that the drilling force is minimal when drilling the seed kernel. When drilling the seed shell, as the drilling depth increases, the friction between the hole wall and the guide part of the drill increases owing to the vibration and wear of the drill bit, and the drilling force increases. Therefore, the drilling force required to drill the down-shell is larger. In addition, the outer side of the seed shell is a hard outer seed coat and cuticle (Figure 4a), and the inner side is a loose and porous inner seed coat (Figure 4c). Therefore, the outer side of the seed shell is harder than the inner side and the drilling force on the outer side is greater than that on the inner side. Thus, drilling force decreases when drilling into the up-shell and increases when drilling out of the down-shell.

Figure 8 shows the relationship between the drilling speed and the diameters of the inlet and outlet of the holes for a given drill diameter and feed rate. When the feed rate was 0.03 mm/r and the drilling speed was 6–24 m/min, the diameters of the inlet and outlet of the holes varied from 0.8 to 1.4 mm for different drill diameters. This figure also shows that when the drill diameter and feed rate are constant, the diameters of the inlet and outlet of the holes increase with increasing drilling speed. This is because a low drilling speed is accompanied by a larger drilling force and vibration amplitude of the drill bit, which results in the rupture of the wood-fiber structure of the seed shell, more burrs in the drilling process, and a lower hole-processing accuracy [27]. These factors reduce the diameter of the holes. As the drilling speed increases, the drilling force gradually decreases and the drill bit is less likely to vibrate. This improves the accuracy of hole processing, results in the hole diameter being closer to its ideal value, and increases the diameters of the inlet and outlet of the holes. However, drills with smaller diameters bend and deform when drilling because of their low stiffness and the hard structure of the *S. mukorossi* seeds. This deformation increases with drilling speed [28], causing the diameter of the drilled hole to increase and deviate from its ideal value. This situation is shown in Figure 8 for a 0.9 mm drill bit.

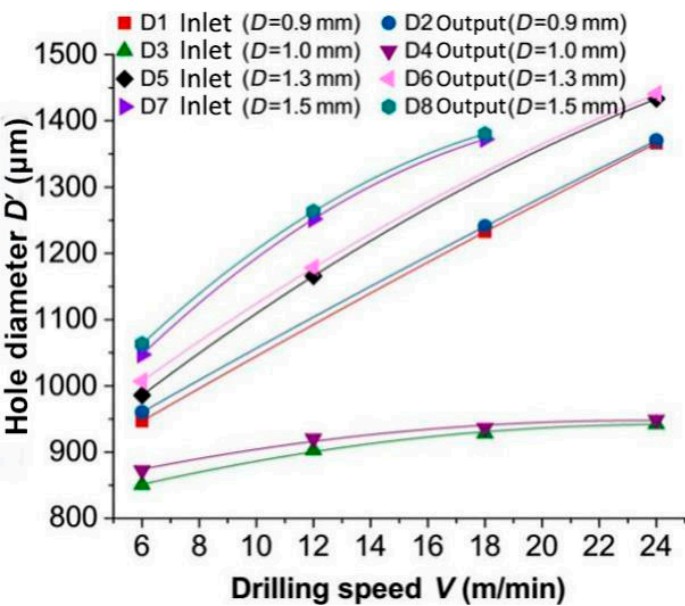

**Figure 8.** Hole diameter as a function of drilling speed. The relationship between *V* and *D′* when *D* = 0.9, 1.0, 1.3, and 1.5 mm, and *U* = 0.03 mm/r.

This analysis demonstrates that to achieve good drilling results, improve the quality and accuracy of the drilling process, and make the hole diameter close to the ideal value, the drilling speed should be as high as possible. However, an increase in drilling speed will accelerate the wear and tear of the drilling tools and reduce their durability. Moreover, small-diameter drills are prone to excessive bending and deformation at high drilling speeds. Therefore, a smaller drilling speed should be selected to improve drill bit durability under the condition of ensuring the quality of processing, and the recommended drilling speed is 6 m/min.

### 3.4. Effect of Drill Diameter on the Drilling Force

Figure 9a,b show the relationship between the drill diameter and the drilling forces generated at different drilling positions in the hole for a given feed rate and drilling speed. The drilling force was up to 15 N among the different positions under the experimental conditions of a feed rate of 0.03 mm/r, a drilling speed of 6 m/min, and drill diameters of 0.9, 1.0, 1.3, and 1.5 mm. The drilling forces at each position were relatively low when the recommended feed rate (0.03 mm/r) and drilling speed (6 m/min) were selected, using drills of 0.9 and 1.5 mm diameter. There are several reasons for this. First, when the drill diameter is smaller (i.e., the depth of the cut is smaller), the cutting width and cutting area are smaller, which lowers the drilling force. In addition, a smaller drill diameter is conducive to drilling the wood-fiber structure of the seeds, improving the accuracy of the drilling process [29], and reducing the friction between the hole wall and the guiding part of the drill, which all serve to reduce the force. Furthermore, a smaller drill diameter reduces the amount of removed material, the drilling power and the generation of drilling heat, helping to slow the wear of the drill bit, thus improving drilling efficiency and reducing the drilling force. Therefore, when using a drill bit of 0.9 mm diameter, the drilling force is relatively small. However, because of the lower stiffness of drills of smaller diameters, problems such as drill vibration and bending deformation can arise, resulting in increased drilling forces and reduced hole-processing accuracy [30]. Thus, increasing the diameter of the drill, to a certain extent, makes stable drilling easy and reduces the drilling forces. Therefore, when using the drill bit of 1.5 mm diameter, drilling is more stable and the drilling force is relatively small.

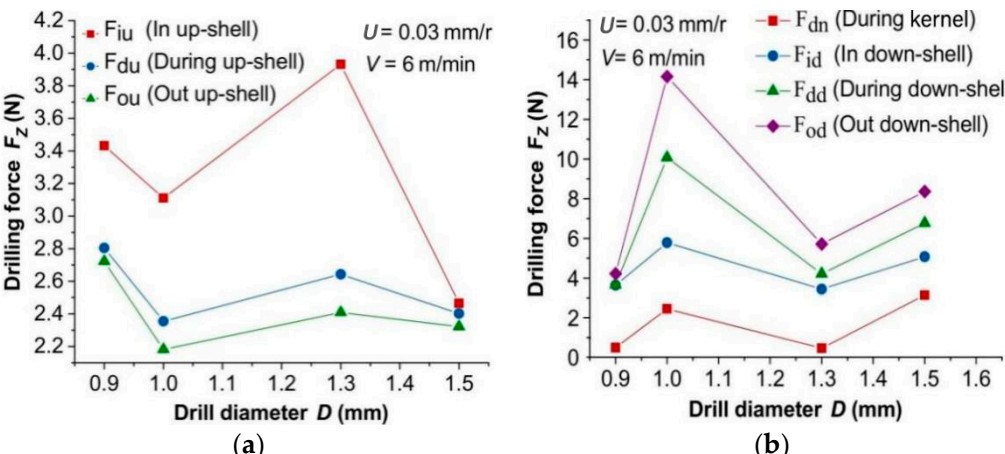

**Figure 9.** Drilling force as a function of drill diameter. (**a**,**b**) The relationship between $D$ and $F_z$ when $U = 0.03$ mm/r and $V = 6$ m/min.

Based on these analyses, it is recommended to use 0.9 mm and 1.5 mm diameter bits to meet machining requirements and improve drilling stability and accuracy.

### 3.5. Effect of Drilling on Hole Morphology

The inlet and exit morphologies of the hole drilled with a 0.9 mm diameter drill, at a feed rate of 0.03 mm/r and a drilling speed of 6 m/min, are shown in Figure 10 for different *S. mukorossi* seeds. The hole–inlet morphologies are relatively complete, but there are a few burrs on the edges (Figure 10a). This is because the vibration during the rotation of the drill bit caused the sub-rear face of the guide section to scratch the hole wall, creating a small number of burrs at the hole inlet [31]. However, the integrity of the hole–exit morphologies is poor, with visible burrs and a substantial loss of roundness in the hole (Figure 10b). This is because the outer part of the seed shell is harder, and the drilling force at the outlet is larger when drilling the seed. The drill bit is impacted at the moment of drilling out the seed, resulting in part of the shell structure breaking and adhering to the edge of the hole, and there were obvious burrs at the outlet of the hole. In addition, as the drilling depth increases, the drilling force increases, making it easy to bend and deform the drill bit [32] and deviate from the original axis, resulting in a more serious loss of roundness at the hole exit.

To address the problem of burrs on the hole wall when drilling *S. mukorossi* seeds and to thus improve the accuracy, the drilling depth can be increased after drilling through the seeds and the burrs can be scraped off by the auxiliary cutting edge of the guide part of the drill. To minimize the loss of roundness of the hole exit, a stiffer bit can be used to reduce the bending deformation of the drill bit, thereby avoiding skewing of the hole axis. In addition, the use of a more suitable drilling die is conducive to the clamping and positioning of the workpiece (*S. mukorossi* seeds), reducing the vibration of the workpiece during drilling and improving the accuracy of the hole. In addition, for *S. mukorossi* seeds, a combination of drilling and reaming can be used to improve the accuracy of the position and shape of the holes.

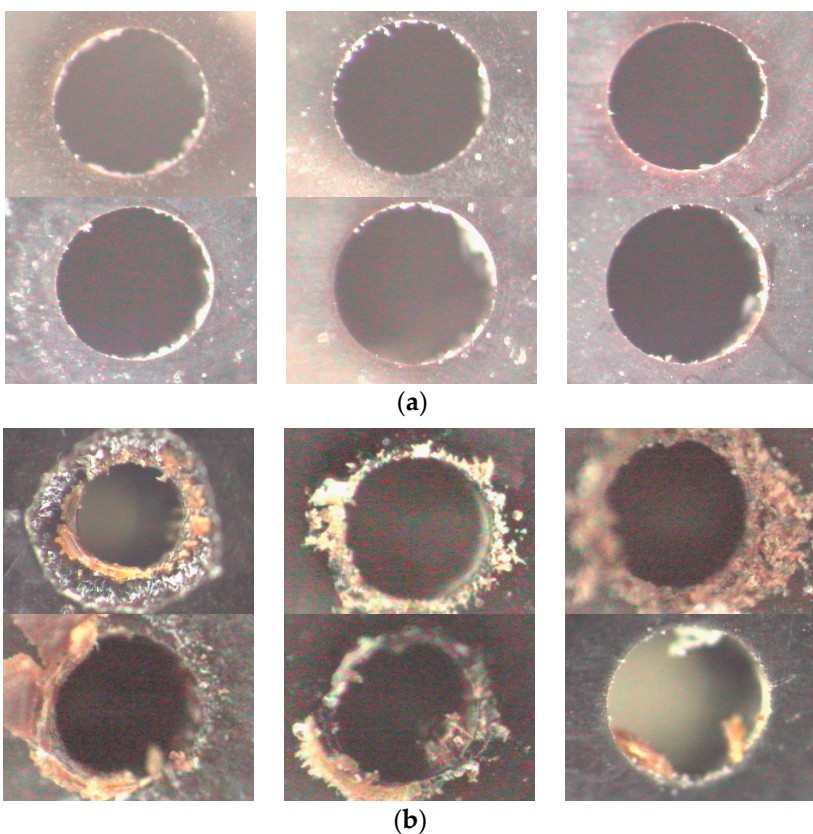

**Figure 10.** Hole morphology of the inlet and outlet for different *S. mukorossi* seeds. (**a**) Inlet morphologies; (**b**) Outlet morphologies.

## 4. Conclusions

This study methodologically reports the effects of drilling parameters (feed rate, drilling speed, and drill diameter) on the drilling force, hole diameter and hole morphology of micro-drilled *S. mukorossi* seeds. A smaller feed rate was found to be conducive to reducing the drilling force and avoiding problems such as low hole accuracy and high surface roughness due to drill lead-off and workpiece vibration. Appropriately increasing the feed rate is beneficial in shortening drilling time and increasing productivity. Considering these factors, a feed rate of 0.03 mm/r is recommended. A higher drilling speed leads to a higher quality and accuracy of the drilling process, resulting in the hole diameter being closer to its ideal value. By contrast, a lower drilling speed helps reduce the wear of the drilling tool and prevents excessive bending and deformation of the small diameter drill bit, thus improving productivity and ensuring the quality of the process. Considering these factors, a drilling speed of 6 m/min is recommended. A smaller drill diameter reduces the drilling force and drilling power, whereas a larger diameter helps avoid bending and deformation of the drill bit and ensures drilling stability. When the ideal diameter of the hole to be machined is small, a drill bit with a diameter of 0.9 mm is recommended, and when the ideal diameter of the hole to be machined is large, a drill bit with a diameter of 1.5 mm is recommended.

**Author Contributions:** Conceptualization, S.Z. and X.B.; methodology, S.Z. and D.X.; validation, S.Z. and X.B.; formal analysis, S.Z. and X.B.; investigation, S.Z. and W.C.; writing the original draft, S.Z. and W.C.; reviewing and editing the manuscript, S.Z. and X.B.; supervision, X.B. and D.X. All authors have read and agreed to the published version of the manuscript.

**Funding:** This research was funded by the National Natural Science Foundation of China (52206229), and National Key Research and Development Program (2019YFD1002401). And The APC was funded by the Fundamental Research Funds for the Central Universities (BLX202127).

**Data Availability Statement:** Test methods and data are available from the authors upon request.

**Conflicts of Interest:** The authors declare no conflict of interest.

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
