# Peer review of "Effects and Mechanism of Micro-Drilling Parameters on the Drilling Force and Hole Morphology of Sapindus mukorossi Seeds"

_forests, doi:10.3390/f14061162_

Round 1
Reviewer 1 Report (Previous Reviewer 1)
The authors made significant changes that improved the readability and quality of the manuscript.
The text has been partially changed by removing passages. Now the introduction has been improved, it is more consistent.
Moisture information has been added, which is crucial in mechanical testing.
Effect of drilling on pore morphology has been described much better, although it is still only an observation but contains enough information to make conclusions.
Suggested minor errors and inaccuracies have been corrected.
In the introduction, the order of paragraphs was changed and thus the citation numbers are out of sequence, the order of citations in the text should be corrected.
Author Response
Please see the attachment

Reviewer 2 Report (New Reviewer)
Dear Authors,
the work entitled "Effects and mechanism of micro-drilling parameters on the drilling force and pore morphology of Sapindus mukorossi seeds" seems to be an interesting and quite catchy topic. However, I see a few flaws in it, some of which seem quite serious and deserve your more attention:
first of all, I am concerned about the number of tests performed for one set of parameters - one drilling for such diverse material as natural seeds seems to be not very scientific. If more of these attempts were made, it requires a clearer emphasis in the content of the work.
the second thing is the presentation of the results. Due to the large variety of parameters, it is extremely difficult to present all the results, which I am of course aware of. My suggestion is to present the results in the form of a table, while the most interesting ones should also be transferred to the graph, e.g. in the form of the average for all three layers of "drilling" of the first coating or averages of each „drilling parameters” divided by layer. This will make it easier for the reader to reach the same conclusions.
I put more comments on the sent pdf of your work. Please treat them as suggestions.

I am not a native speaker so my comments may not be correct. However, I am sure that the use of the term pores for holes after drilling is misleading.
I put more comments on the sent pdf of your work. Please treat them as suggestions.
Author Response
Please see the attachment

Reviewer 3 Report (New Reviewer)
The manuscript titled “Effects and mechanism of micro-drilling parameters on the drilling force and pore morphology of Sapindus mukorossi seeds” concerns the effects of these drilling parameters (feed rate, drilling speed, and drill diameter) on the drilling force, pore diameter and pore morphology.
The ideas in the manuscript are very interesting, and the research results obtained have some potential applications of machining. This manuscript is well structured and well written, which is easy to follow. The figures and tables are neat and easy to understand. The methodology is thoroughly explained and the work overall seems to be skillfully performed. I believe this work provides valuable information for those with interests in materials processing.
Generally, I think this manuscript can be accepted for publication after minor revision.
1. The range of drilling parameters for the drilling experiments was determined by test drilling. What is the test drilling? Reference.
2. How many times each group of parameter combinations was repeated?
3. Provide all the model for the equipment you used in this work.
4. Why only the Fz was in focus.
5. What is the sampling frequency of cutting force?
6. In the wood machining, U is used to define the feed rate. fz is often adopted in the metal machining.
7. During kernel (Fig. 6b), why the Fz increased first and decreased with the increased of drilling speed. Explain more.
8. Provide the limitation of your work.
Round 2
Reviewer 2 Report (New Reviewer)
Current form of article is good for publication.
Thank You.
This manuscript is a resubmission of an earlier submission. The following is a list of the peer review reports and author responses from that submission.
Round 1
Reviewer 1 Report
The manuscript entitled “Effects of micro-drilling parameters on the drilling force and pore morphology of Sapindus mukorossi seeds” present an interesting, although niche, topic. The authors conducted tests on unusual biological material. They showed different reactions in response to drilling.
1. The introduction discusses the wide-ranging applications of Sapindus mukorossi seeds. The authors also showed some research gaps in micro-drilling technology supporting the need for this work. However, it is not clear why this topic of seed drilling is important. There is no information about the scale of the practice. From the introduction, it seems that this practice has been known for a long time. Why is research needed now? I get the impression that the authors show the need for miro-drilling studies of various materials, and the seeds are only an example. So this is more of a case study of drilling, rather than the validity of seeds.
2. The authors reported general conditions for drying seeds, but for the sake of the essence of a scientific approach, they should give the moisture content of the seeds. Especially since it is quite easy to calculate. The moisture content determines the reaction of the material and is a rather important factor.
3. In Confusions we read
"This study methodically reports the effects of drilling parameters (feed rate, drilling speed, and drill diameter) on the drilling force and pore morphology"
Pore morphology was practically not studied. The image shown in Figure 9 and the corresponding text of the paper are unclear. We couldn’t find of how many photos were taken? Are the images in the Figure 9 a compilation (e.g., an average) obtained from more than one observation? If yes, how was this calculation done? If this is just one example, the picture presented may be completely random. There is a lack of statistical analysis for the various options and scientific basis for conclusions on this topic.
4. Why was a particle-size-analysis microscope used for diameter measurements? After all, this device is used to measure particles.
5. Relevant descriptions need to be checked and corrected:
Caption of Figure 4 descriptions of figures a-d are missing
Caption of Figure 5 Borehole diameter against feed speed(?) – or feed rate (?)
Caption of Figure 8 descriptions of figures a-d are missing
6. The work needs to improve the readability (e.g. shortening long sentences), especially in the results section.
The manuscript file does not have numbered lines, which makes it difficult to plan comments.
Reviewer 2 Report
The work is well conceived, and it has the basis for such research, which is well explained in the introduction. There is room for improvement, such as the overlapping of some directly related terms such as hole (diameter) and pore (morphology) or feed rate and drilling speed, between which there is practically no difference in this research. The research procedure and the equipment used are well described. There is also room for corrections and refinement in the Materials and methods chapter such as the used term "spare parts" in the second sentence of chapter 2.2. which is not clarified.
The problem of drilling such a specific material is also well presented, even a relatively well thought-out way of fixing the samples for drilling, although the main factor of unpredictable factors should have been better defined due to the uneven shape of S. mukorossi seeds. One of the objections refers to the non-use of equipment that could measure force in different directions, instead authors used only one, in the direction of the z-ax.
In addition to all this, a high-quality discussion was conducted on the obtained results, however, the main objection to the entire work refers to the scientific unfoundedness of the conclusions presented. They are more assumptions that rely on observation rather than on the results of measurements, which are not enough for this work to be published as a scientific paper.